# Assessing Climate Change Exposure for the Adaptation of Conservation Management: The Importance of Scale in Mountain Landscapes

**Mónica Gómez-Vadillo †, Mario Mingarro \*,† , Guim Ursul and Robert J. Wilson**

Departamento de Biogeografía y Cambio Global, Museo Nacional de Ciencias Naturales (MNCN-CSIC), 28006 Madrid, Spain
\* Correspondence: mario_mingarro@mncn.csic.es
† These authors contributed equally to this work.

**Abstract:** Vulnerability of mountain ecosystems to climate change depends on the capacity of topographic variation to provide heterogeneous microclimates and rates of climatic change. Accurate methods are therefore needed to assess climate at spatial resolutions relevant to ecological responses and environmental management. Here, we evaluate a mechanistic microclimate model (30 m resolution; Microclima) and mesoclimate data (1 km; CHELSA) against in situ temperatures, finding that both capture (whilst somewhat underestimating) variation well in observed ground-level maxima along a mountain ridge in 2011-13. We apply the models to estimate ecological exposure to recent temperature changes for four mountain areas of the Iberian Peninsula, based on analogous and non-analogous monthly maxima in 1980–1989 versus 2010–2019. The microclimate model revealed fine-resolution exposure to non-analogous conditions that were concealed in mesoclimate data, although whether exposure was greater at the micro- or mesoscale (and hence the types of organisms or management decisions affected) depended on the topographic context of each mountain range. Habitat type influenced microclimatic exposure, and hence may provide opportunities for conservation adaptation. These results suggest that mechanistic models are potentially useful tools to assess exposure to climate change at spatial resolutions that permit understanding and management of biodiversity responses in mountain ecosystems.

**Keywords:** analogous climatic conditions; climate change exposure; mesoclimate; microclimate; mountain ecosystems; refugia; spatial resolution



## 1. Introduction

Mountain ecosystems are a high priority for adapting conservation management to climate change [1]. The rapid warming to which many high elevations have been exposed threatens endemic species that are restricted to rare mountain climates and isolated from locations that will be climatically favourable in the future [2,3]. Mountains at the warm, "rear" edges of geographic ranges may also represent important reservoirs of genetic diversity and adaptive potential because they have acted as refugia by supporting persistent populations of species through glacial and interglacial climates [4]. In these environments, microclimatic and mesoclimatic heterogeneity resulting from mountain topography could buffer species against climate change by providing opportunities for behavioural thermoregulation and fine-scale distribution shifts [5]. However, to establish the capacity of mountains to act as refugia, and to prioritize additional management or designation of protected sites to adapt their conservation, it is necessary to assess variation in climatic conditions and rates of change at spatial scales that are relevant to the "valued resources" (e.g., species or communities) that are threatened by ongoing climate change [6].

Until now, many assessments of observed and predicted species range shifts in response to climate change have used coarse-scale (approximately 10–100 km) macroclimatic

data, over broad geographic gradients such as latitude [7–9]. In reality, species experience variation in climate at much finer spatial resolutions across their geographic ranges [10]. Mesoclimate (approximately 100 m–10 km) varies because of factors such as elevation, topography, distance to the coast, and the weather patterns that these create [11–14], as well as effects of land cover on shading and wind exposure [15,16]. Microclimate (<100 m, or much finer resolutions depending on the size of the organism) is influenced by very fine-scale variation in topography and land cover, and may have important effects on organism performance and population dynamics [17–19].

Both macroclimate and mesoclimate are often determined with the same methodology, by interpolations of weather station data using patterns such as lapse rates. Studies that have sought to understand the ecological influence of climatic variation (e.g., [20,21]) have relied on data from national networks or online databases such as Worldclim [22] or CHELSA [23,24]. At regional scales of mountain ranges, elevation is the main factor determining temperature and precipitation, with the coldest and wettest climatic conditions found at higher elevations [11,13]. However, cool and moist conditions can occur in valley bottoms because of factors such as cold air pooling and hydrological accumulation, whilst variation in topography and orientation creates isolated microclimates by interacting with the physical processes of air flow and solar radiation [6,14,24,25]. These factors have important effects on the responses of populations and communities to climate change: for example, butterfly community composition in the Iberian Peninsula has responded less to climatic warming and drying since 1980 in locations where greater topographic heterogeneity has reduced rates of climate change or permitted fine-scale species redistributions [26].

Accurate mesoclimatic and microclimatic data may therefore be important when assessing ecological vulnerability to climate change in mountains and when prioritizing locations or habitats for management or protection. In this context, interpolated data may not be representative: partly because there are very few meteorological stations in high mountains in most parts of the world [27]; and partly because complex topography causes local temperature to vary substantially, resulting in mosaics of mesoclimatic and microclimatic conditions that differ considerably from the macroclimate [19]. Recent advances in mechanistic modelling now make it possible to estimate microclimate based on topography and vegetation cover at the scales relevant to organismal exposure [28,29]. These models can be applied to any location at any time, and field validation shows that they can provide accurate estimation of microclimatic conditions in isolated and topographically diverse environments [30].

In this paper, we first evaluate the accuracy of 1 km resolution mesoclimate data (CHELSA; [23]) and a 30 m resolution microclimate model (Microclima; [28]) using ground-level field measurements of temperature from a mountain ridge in 2011-13 [31]. We then apply modelled mesoclimate and microclimate data to test the effects of spatial scale on recent exposure to climate change in four mountain regions of central Spain. We take an approach based on the occurrence of analogous versus non-analogous climatic conditions [2] between two time periods (1980-89 and 2010-19). We define analogous conditions as those that were present in both periods in a reference grid square, and non-analogous conditions as those that were only present in the square during one of the two time periods (conditions either lost or gained from one period to the other). We calculated analogous and non-analogous conditions based on monthly temperature maxima across grid squares centred on the mountain ranges: 6 × 6 km grid squares with microclimate modelled at 30 m resolution; and 50 × 50 km squares with mesoclimate data at 1 km resolution. We use these spatial extents and resolutions to represent different scales of ecological observation or conservation management, e.g., ranging from an individual population or metapopulation of a rare plant or insect (6 × 6 km square), through to a system of insect metapopulations, or the home ranges of large vertebrates, or birds undertaking elevation migrations (50 × 50 km square) (e.g., see [32] for an example of the spatial scales at which a mountain mammal responds to climate change). We also use the microclimate model to estimate differences in exposure to non-analogous conditions

in three different habitats typical of the study regions (grassland, scrub and coniferous woodland). Overall, we seek to validate the use of a mechanistic microclimate model for Mediterranean mountain habitats, and to demonstrate how the scale dependence of exposure to changing climates influences the capacity of mountains to act as refugia in adapting conservation to climate change.

## 2. Materials and Methods

### 2.1. Study Area

The study included four Mediterranean mountain ranges at a similar latitude in the central Iberian Peninsula, two in the Sistema Central (Sierras de Guadarrama and Gredos) and two in the Sistema Ibérico (Sierras de Albarracín and Javalambre) (Figure 1). The research was part of a project to understand the capacity of mountains to act as climate change refugia, focused on these four regions because each has recent historical (1985–2005) field data on butterfly distributions over elevation gradients [33,34]. The vulnerability to climate change of biota in these locations is important for conservation because they support many endemic species and species at their rear range margins: these species are expected to suffer decreasing regional distribution sizes or extirpations as conditions warm [35,36], partly because all four mountain ranges show decreases in area as elevation increases [37]. To conduct our comparisons of mesoclimatic and microclimatic change, we selected focal locations where historical butterfly sample sites were present at a comparable average elevation in the centre of each system (1300–1500 m above sea level). These acted as the focal grid cells (6 × 6 km) for microclimatic modelling in each system. We delimited 50 × 50 km grid cells around the centroid of each 6 × 6 km grid square to conduct our mesoclimate analyses (Figure 1).

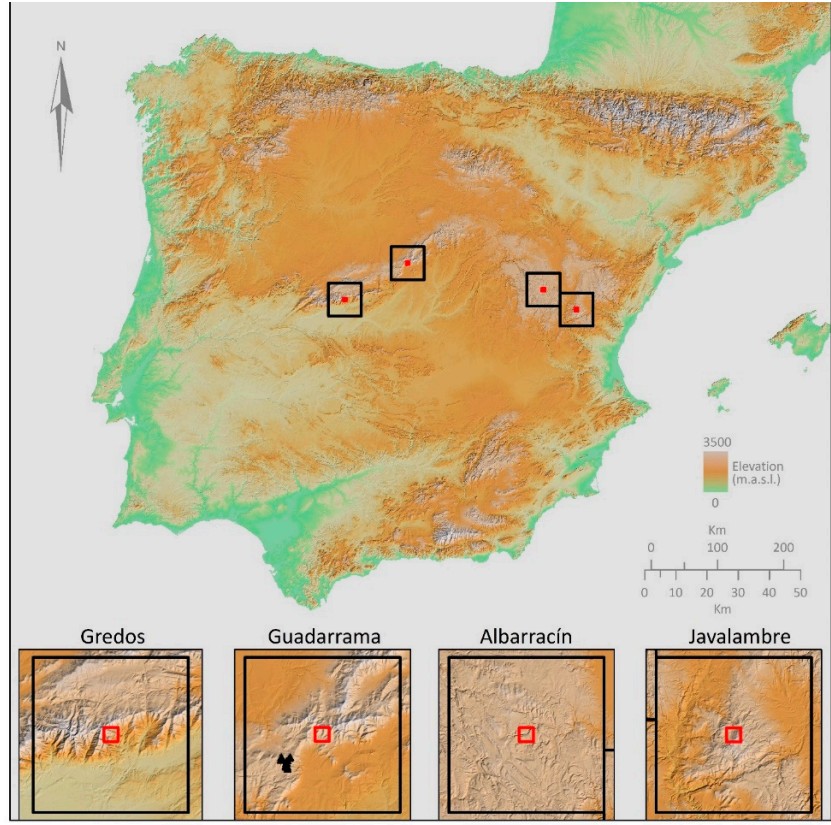

**Figure 1.** Location of the four mountain regions in the Iberian Peninsula. Black polygons represent 50 × 50 km squares for mesoclimate analysis, centred on red 6 × 6 km squares for microclimate analysis. Guadarrama inset: black triangles show locations of dataloggers for field validation of temperatures.

### 2.2. Mesoclimate and Microclimate Data

We estimated mesoclimatic conditions using the "Climatologies at high resolution for the earth's land surface areas" (CHELSA) dataset (www.chelsa-climate.org, accessed date 23 January 2021) [23]. CHELSA downscales ERA-Interim climate data from the European Centre for Medium-Range Weather Forecast, using temperature interpolations in mountain regions based primarily on altitude. For our analyses, we extracted the monthly minimum and maximum temperatures (°C) at 1 km resolution for our study locations from version 2.1.

Microclimate modelling was carried out in the R package Microclima [28] using the runauto function to integrate the mechanistic physical models Microclima and NicheMapR [38]. These models allow for the estimation of hourly temperatures at fine spatial resolutions based on freely available high-temporal-resolution climate data [29]. In order to model the effects of topography and/or vegetation on convection and radiation, and hence the temperatures experienced near the ground, NicheMapR provides a vertical-flow air and soil microclimate model, and with the resulting data, Microclima calculates the effect of physical forcing on near-ground temperature. In our analysis, soil and canopy albedo were set to 0.15 and 0.23 as default parameters. In Microclima, habitat type determines leaf area, and hence shading at ground level. We modelled microclimate based on the three habitats with the highest coverage in the four 50 × 50 km cells: evergreen needle leaf forest "H1", open scrubland "H7" and short grassland "H10" [38] (see Figure 2). Coastal effects were not implemented in the microclimate models to reduce computational time and because the systems are continental mountains distributed at least 50–250 km from the coast. We estimated hourly temperatures 10 cm above the ground at 30 m resolution, and then calculated average monthly daily minimum and maximum temperatures to compare with the mesoclimatic (CHELSA) and datalogger temperatures.

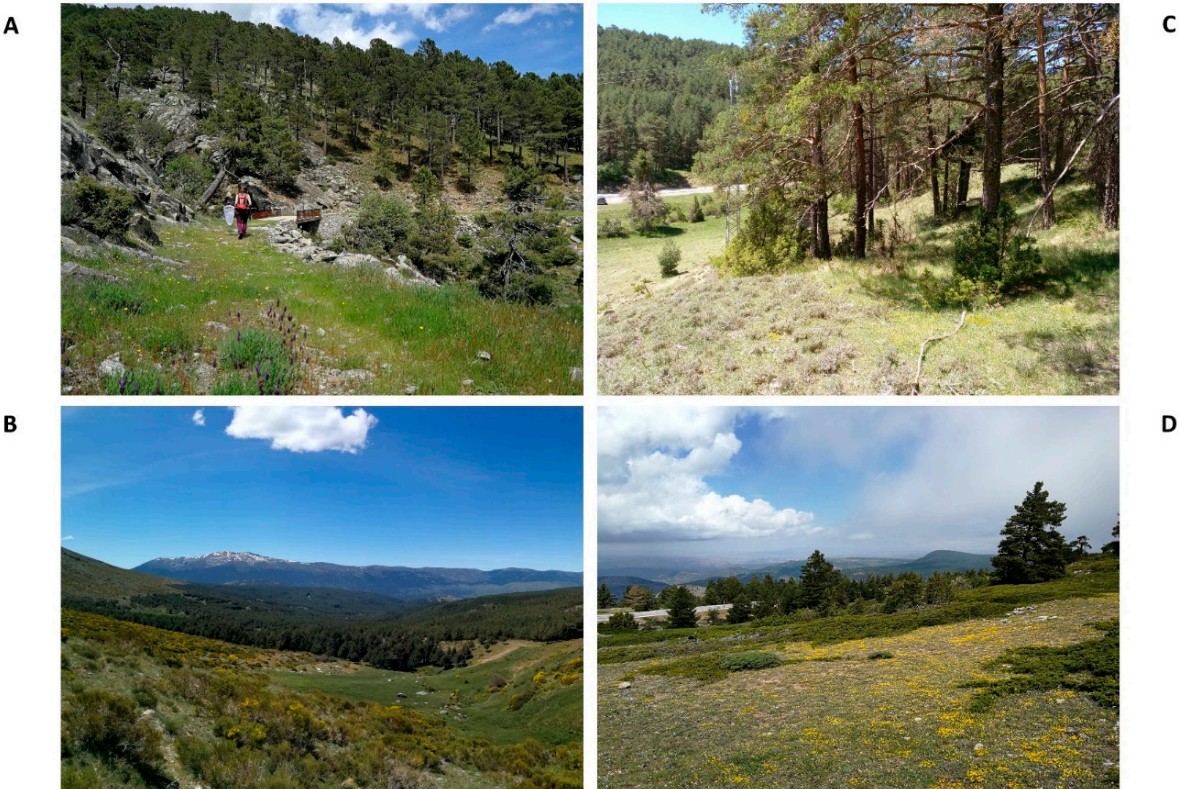

**Figure 2.** Photographs of habitats in the four mountain study regions. (**A**) Sierra de Gredos, (**B**) Sierra de Guadarrama, (**C**) Sierra de Albarracín, (**D**) Sierra de Javalambre. The main habitat types are: H1—pine forest; H7—open scrubland; H10—short grassland.

### 2.3. Field Validation of Mesoclimate and Microclimate

To evaluate the accuracy of the mesoclimate and microclimate models for mountains in the central Iberian Peninsula, we used temperature data from Hobo Tidbit dataloggers from a 4 × 3 km ridge in the Sierra de Guadarrama from October 2011 to June 2013 (Figure 1; see [31]). These loggers had been used to record effects of topography on the temperatures experienced by larvae of the alpine grassland butterfly *Parnassius apollo* [31,39]. Twenty-five loggers were placed at elevations of 1570–1890 m a.s.l. at ground level in the shade of dwarf shrubs (5–10 cm vegetation height; mean 7.5 cm). In our previous study, spring temperatures (during the period of larval growth) recorded by the loggers were negatively related to elevation and positively related to modelled solar insolation (depending primarily on aspect) [31]. To evaluate the accuracy of mesoclimate and microclimate models, we calculated monthly minima and maxima (°C) from the hourly temperatures recorded by the dataloggers. Because of loss or damage to some of the 25 dataloggers over the course of 21 months, the total sample size was 525 monthly temperatures (Supplementary Materials).

Georeferences of the dataloggers were used to determine the 30 m or 1 km resolution cell for comparison with the microclimate or mesoclimate models, respectively. To perform comparisons between the data sets, Pearson correlations were performed and the root-mean-squared error (RMSE) was calculated, following [30] (sample size for all analyses = 525 datalogger months). The RMSE index quantifies the difference between a set of predicted and observed values, indicating the absolute fit to the data, with lower RMSE values representing closer fit between the predicted and observed values [30].

### 2.4. Exposure to Analogous and Non-Analogous Conditions

Reference periods for recent climate change exposure were 1980-89 and 2010-19. Temperatures in the Iberian Peninsula have risen markedly since 1980 [26], and we used 1980-89 as our baseline period (i) because the earliest historical butterfly samples in the mountain field sites assessed by our refugia project were from the 1980s; and (ii) because the latest version of 1 km resolution temperatures (CHELSA) was available from 1979 onwards [23]. We used the twelve months of the year as units of analysis for exposure to analogous and non-analogous conditions, because exposure during specific parts of annual life cycles determines phenology and abundance responses (e.g., [40–42]). We conduct the analyses using monthly maximum temperatures, relevant for growing season length and upper tolerance limits for species at low-latitude or low-elevation range limits. For mesoclimate, we extracted monthly maximum temperatures at 1 km resolution throughout each of the four 50 × 50 km squares, using CHELSA data from 1980-89 to 2010-19 [23]. For microclimate, we calculated monthly maxima by modelling hourly temperatures at 30 m resolution throughout each of the four 6 × 6 km squares for 1980-89 and 2010-19, for each of the three reference habitats representing a gradient from open (H10) through intermediate (H7) to closed vegetation (H1) (Figure 2).

Thus, for 1980-89 we calculated the range of mean daily maxima for each month (January, February etc.) in each pixel of the focal squares (30 m pixels in 6 km squares; 1 km pixels in 50 km squares). This gave the absolute minimum and maximum values of maximum monthly temperature recorded anywhere in the square in the decade at the spatial scale of study. We also calculated the full range of monthly maxima for 2010-19. It was then possible to calculate for each month of the year whether each pixel showed analogous or non-analogous conditions between the two decades. Pixels with "analogous" temperatures were those whose temperature in 1980-89 fell within the full range of monthly maximum temperatures in any pixel in the square for that month in 2010-19, and whose 2010-19 temperature was likewise present across the square in 1980-89. There were two types of pixel with "non-analogous" temperatures: "lost", where the temperature in 1980-89 did not occur anywhere in the square in 2010-19; and "gained", where the 2010-19 temperature had not occurred anywhere in 1980-89 (Figure 3). We therefore calculated the total proportion of pixels that were "analogous" versus "non-analogous" (lost + gained) for each month, in each square, and at each spatial scale (Supplementary Materials).

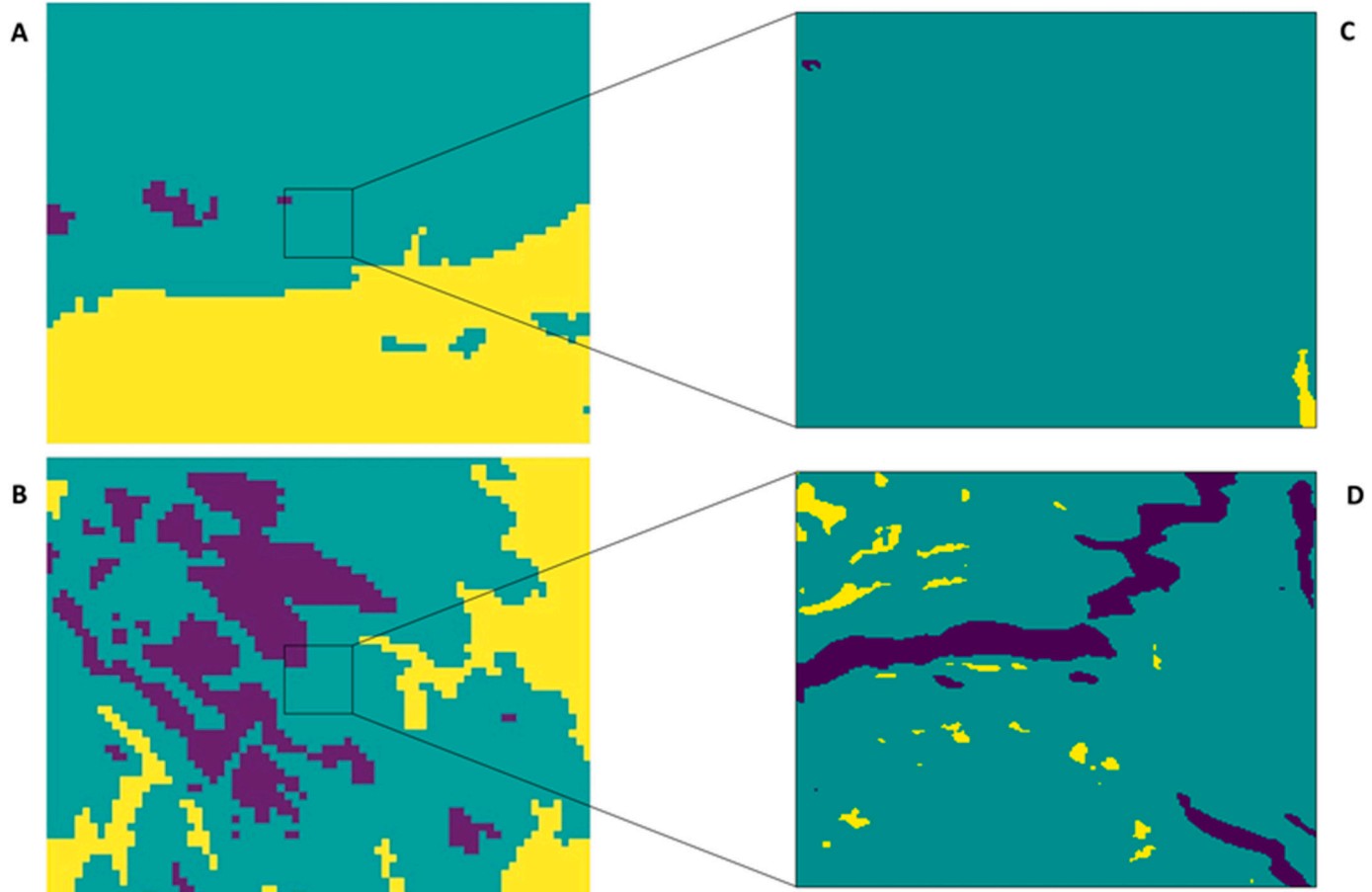

**Figure 3.** Exposure to analogous and non-analogous maximum July temperatures from 1980-89 to 2010-19 in the Sierra de Gredos (**A**,**C**) and Sierra de Albarracín (**B**,**D**). Green represents analogous conditions, while yellow indicates non-analogous "gained" conditions and purple represents non-analogous "lost" conditions. (**A**,**B**): Mesoscale 50 km squares, 1 km resolution; (**C**,**D**): Microscale 6 km squares, 30 m resolution, modelled for H10—grassland.

*2.5. Statistical Analysis*

We sought to test whether the scale of observation affected exposure to non-analogous temperatures for mountain taxa, and whether exposure differed among the four mountain regions or in different seasons of the year. We conducted generalized linear models (GLM) with the proportion of pixels showing analogous conditions as the response, using quasi-binomial error because data were bounded between 0 and 1 and the relationship between residual deviation and residual degrees of freedom indicated overdispersion [43]. The independent variables (all categorical) were scale (microclimate or mesoclimate), region (Gredos, Guadarrama, Albarracín or Javalambre) and season. In initial models, we grouped the months as winter (December–February), spring (March–May), summer (June–August) and autumn (September–November), but found that model performance was improved by grouping the autumn and winter months together: the results presented therefore include season as a three-factor variable (combining spring and summer months did not improve the models). We also tested for interactions: between scale and region, to assess whether effects of scale on exposure to non-analogous conditions were consistent among the mountain ranges; between scale and season; and between season and region. We calculated all possible models based on the inclusion and exclusion of the explanatory variables and interactions, and ranked these using the quasi-AIC criterion (QAIC). We selected the most parsimonious model based on the lowest QAIC, also considering models

that were within two QAIC units of the best model as well-supported if they included fewer explanatory terms [44].

For the main comparisons of mesoclimatic and microclimatic exposure to non-analogous conditions, we present results using microclimate modelled for two of the reference habitats: H10 (short grassland) and H1 (pine forest). We conducted an additional GLM to test the effects of habitat type on microclimatic exposure. In this case, the explanatory variables were habitat type (pine forest "H1", open scrubland "H7" and grassland "H10"), region, season and the respective interaction terms.

## 3. Results

### 3.1. Field Validation of Mesoclimate and Microclimate Data

For monthly maxima, both the 1 km resolution mesoclimate data and the 30 m resolution microclimate model were closely correlated with field measurements of temperature ($n$ = 525; r = 0.89 and 0.86, respectively) showing that the relative variation over time and among the different datalogger locations was well-captured by both models (Figure 4). However, the temperature range of monthly maxima was much greater for the field measurements (minimum −1.4, maximum 63.4 °C), particularly during summer months (June–September) when there was little overlap of field-recorded monthly maxima with the microclimate model, or, especially, the mesoclimate data (Figure 4). Nevertheless, the microclimate model captured a greater overall range of monthly maxima (total range 30.5 °C, minimum 2.1, maximum 32.6) than the mesoclimate data (23.7 °C, 3.1–26.8), and the absolute values of the field measurements were marginally closer to the microclimate model (RMSE = 8.17) than to the mesoclimate data (RMSE = 9.74). The modelled microclimate and mesoclimate data were very closely correlated (r = 0.96) and had very similar absolute values (RMSE = 3.65).

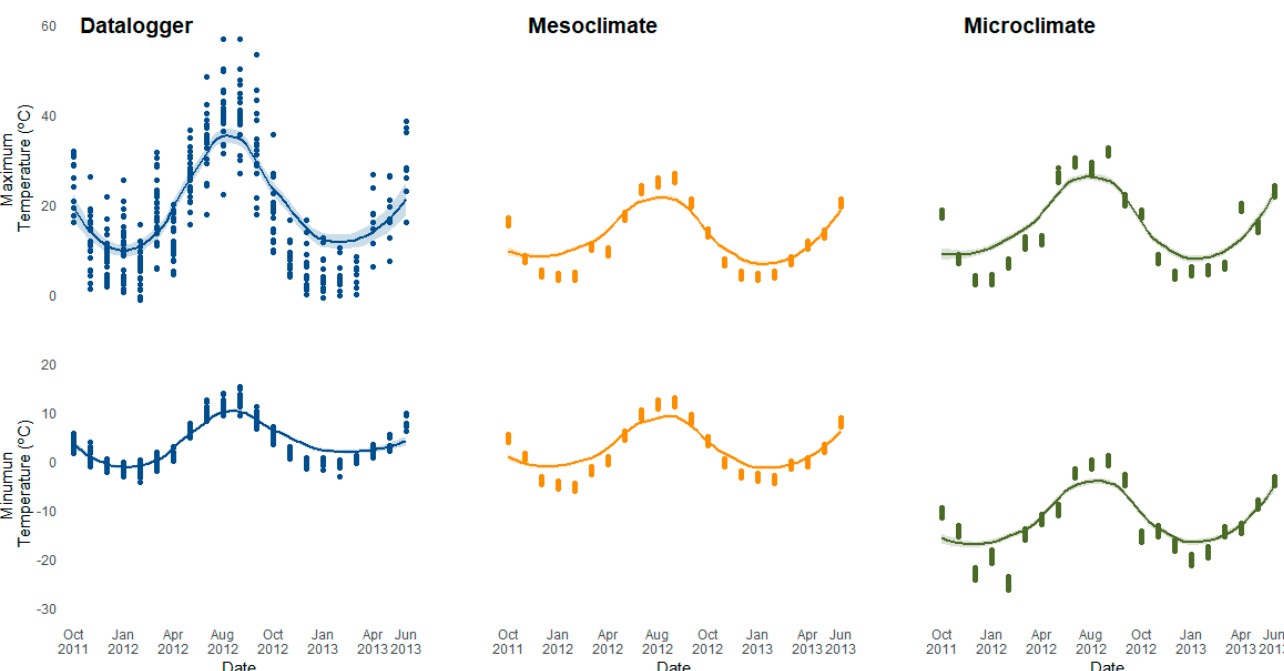

**Figure 4.** Field validation of monthly maximum (above) and minimum (below) temperatures (°C) in the Sierra de Guadarrama. Temperatures from October 2011 to June 2013 are shown for the 25 dataloggers (left column, blue points), associated 1 km resolution mesoclimate data (centre, orange) and 30 m resolution microclimate data (right, green). Lines show loess curves.

Field-recorded monthly minima were again closely correlated with the mesoclimate data (r = 0.95) and microclimate model (r = 0.90) (Figure 4). However, the absolute values of modelled microclimate temperatures had a broader range and were much lower (minimum

−26.3, maximum 1.3 °C) than the mesoclimate data (minimum −6.0, maximum 13.1) and field measurements (minimum −4.1, maximum 15.2). Consequently, fit with the field minimum temperatures was much closer for mesoclimate data (RMSE = 2.2) than for the microclimate model (RMSE = 16.4).

### 3.2. Exposure to Non-Analogous Temperatures

Given the better fit of the microclimate model to field-measured temperature maxima, analyses of analogous and non-analogous conditions are based on monthly maxima. We present results for microclimate models based on representative closed (H1—pine forest) and open (H10—short grassland) habitats.

The proportion of 1 km squares in the 50 × 50 km regions that maintained analogous monthly maxima from 1980-89 to 2010-19 (mesoclimate analysis) varied from 50–100% (Figure 5). Variation among months in exposure was widest in Albarracín (50–100% analogous) and Gredos (59–97%), and much less in Guadarrama (79–100% analogous) and Javalambre (86–100%).

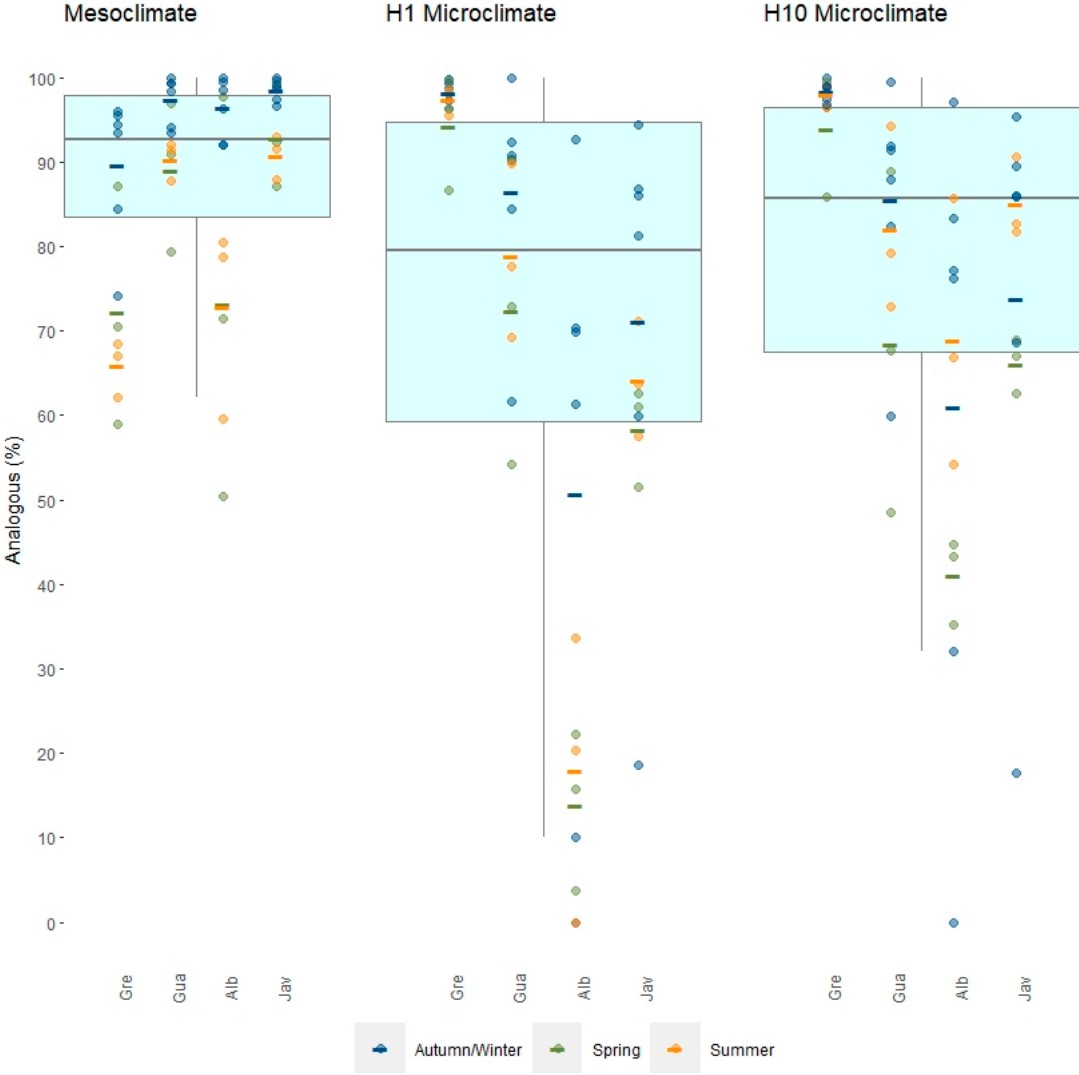

**Figure 5.** The percentage of pixels showing analogous temperatures between 1980–1989 and 2010–2019. Mesoclimate data—1 km resolution in 50 × 50 km cells; microclimate models—30 m resolution in 6 × 6 km cells for H1—pine forest and H10—grassland. Points show % analogous monthly maxima for the four regions; coloured bars show means per season in each region. Box plots show median and interquartile range for all data at meso- or microscales for the respective habitats.

In general, the microclimate analysis using 30 m pixels in 6 × 6 km focal squares was able to detect areas of non-analogous monthly maxima (both lost and gained conditions) that were missed by the mesoclimate data (Figure 4). The proportion of pixels that maintained analogous maxima varied markedly among the regions. Exposure was least in Gredos, where monthly analogous maxima ranged from 85–100%. Monthly variation in exposure was much wider in the other three regions, with analogous conditions ranging from 0–96% in Albarracín, 20–95% in Javalambre, and 47–100% in Guadarrama, depending on month and habitat type (Figure 5).

The most parsimonious models for variation in monthly analogous conditions were generally consistent whether using the microclimate model based on H1 (pine forest) or H10 (short grassland) (Table 1). The selected models included effects of scale, region, season (spring, summer and autumn/winter) and the interaction between scale and region. Overall, exposure was greatest in Albarracín (most negative coefficient), and least in Gredos (reference category: all other regions had negative coefficients) where exposure at the microscale was minimal. Exposure to non-analogous conditions was greater for the microclimate analysis (Figure 5), but with an interaction with region (positive coefficients with Guadarrama, Albarracín and Javalambre, because exposure at the mesoscale was greater in Gredos). The region:scale interaction was not present in the best model if Gredos was removed from the analysis.

**Table 1.** Generalized linear models for the proportion of analogous monthly maximum temperatures between 1980-89 and 2010-19. Reference categories are Region—Gredos; Scale—Microclimate; Season—Autumn/Winter. The best model was selected based on a quasibinomial error term using quasi Akaike information criterion (QAIC). For H10, QAIC of null model = 166.49, QAIC of model shown = 104.40, QAIC of next-best model = 105.50 (including an additional Region:Season interaction). For H1, QAIC of null model = 121.7, QAIC of model shown = 103.6, QAIC of next-best model = 105.4.

| Model | Residual Deviance | Residual DF | Deviance | DF | Factor | Coefficient (±S.E.) |
|---|---|---|---|---|---|---|
| **H1—Pine forest** | | | | | | |
| Null | 979,888 | 95 | | | | |
| Region | 501,952 | 92 | 477,935 | 3 | Intercept<br>Albarracín<br>Guadarrama<br>Javalambre | 4.07 *** (±0.53)<br>−4.32 *** (±0.55)<br>−2.05 *** (±0.56)<br>−2.87 *** (±0.54) |
| Scale | 457,320 | 91 | 44,633 | 1 | Mesoclimate | −2.16 * (±0.84) |
| Season | 383,453 | 89 | 73,867 | 2 | Spring<br>Summer | −1.12 *** (±0.27)<br>−0.86 ** (±0.27) |
| Region:Scale | 326,851 | 86 | 56,602 | 3 | Albarracín<br>Guadarrama<br>Javalambre | 4.70 *** (±1.15)<br>3.41 * (±1.41)<br>4.55 ** (±1.53) |
| **H10—Short grassland** | | | | | | |
| Null | 626,672 | 95 | | | | |
| Region | 416,599 | 92 | 210,073 | 3 | Intercept<br>Albarracín<br>Guadarrama<br>Javalambre | 3.74 *** (±0.54)<br>−3.26 *** (±0.55)<br>−2.14 *** (±0.57)<br>−2.48 *** (±0.56) |
| Scale | 401,853 | 91 | 14,746 | 1 | Mesoclimate | −2.21 * (±0.85) |
| Season | 358,896 | 89 | 42,957 | 2 | Spring<br>Summer | −0.77 ** (±0.25)<br>0.15 n.s. (±0.28) |
| Region:Scale | 317,410 | 86 | 41,486 | 3 | Albarracín<br>Guadarrama<br>Javalambre | 3.64 ** (±1.15)<br>3.49 * (±1.41)<br>4.14 ** (±1.54) |

Significance of coefficients: *** $p < 0.001$, ** $p < 0.01$, * $p < 0.05$, n.s. = not significant, $p > 0.10$.

It is notable that temperatures decreased for several spring and summer months. For both habitat types, exposure to non-analogous conditions was greatest in spring months (March–May). For habitat H1 (forest), summer months also showed greater exposure to non-analogous temperatures than autumn/winter. The deviance explained by the models (Pseudo $R^2$) ranged from 49% for H10 (grassland) to 66% for H1 (pine forest), suggesting that the effects of scale, region and season were captured more clearly for microclimate modelled using habitat H1.

We added an additional intermediate habitat type (H7—open scrubland) for our comparison of the effects of habitat on modelled microclimatic exposure. The differences in exposure among habitats were not sufficient to overcome the differences in microclimatic exposure among regions: irrespective of habitat type, the proportion of analogous conditions was greatest in Gredos, followed by Guadarrama, Javalambre and then Albarracín (Figure 6). For the two regions in the Sistema Ibérico (Javalambre and Albarracín), in which fewer pixels maintained analogous conditions, exposure was greatest for H1 (pine forest), intermediate for H7 (scrub) and least for H10 (grassland). The most parsimonious model for variation in analogous monthly maxima at the microscale included effects of region, habitat type (least exposure for H10—grassland) and season (greatest exposure in spring) (Table 2). There were no interactions of region with habitat or season in the final model, which explained a comparable proportion of deviance to the models comparing microclimatic and mesoclimatic data (Pseudo $R^2$ = 0.55).

**Table 2.** Generalized linear model for the proportion of analogous monthly maximum temperatures between 1980-89 and 2010-19 for modelled microclimate in three reference habitat types: H1 (pine forest), H7 (open scrubland), H10 (short grassland). Reference categories are Region—Gredos; Habitat—H1—pine forest; Season—Autumn/Winter. The best model was selected based on a quasibinomial error term using quasi Akaike information criterion (QAIC). QAIC of null model = 281.55, QAIC of model shown = 142.39, QAIC of next-best model = 144.14.

| Model | Residual Deviance | Residual DF | Deviance | DF | Factor | Coefficient ($\pm$S.E.) |
|---|---|---|---|---|---|---|
| Null | 2,158,894 | 143 | | | | |
| Region | 1,135,085 | 140 | 1,023,809 | 3 | Intercept Guadarrama Albarracín Javalambre | 3.56 *** ($\pm$0.44) −2.12 *** ($\pm$0.45) −3.67 *** ($\pm$0.44) −2.65 *** ($\pm$0.44) |
| Habitat | 1,091,680 | 138 | 43,405 | 2 | Short grassland Open scrubland | 0.52 * ($\pm$0.22) 0.41 † ($\pm$0.21) |
| Season | 966,869 | 136 | 124,812 | 2 | Spring Summer | −0.90 *** ($\pm$0.21) −0.21 n.s. ($\pm$0.22) |

Significance of coefficients: *** $p < 0.001$, * $p < 0.05$, † $p < 0.10$, n.s. = not significant, $p > 0.10$.

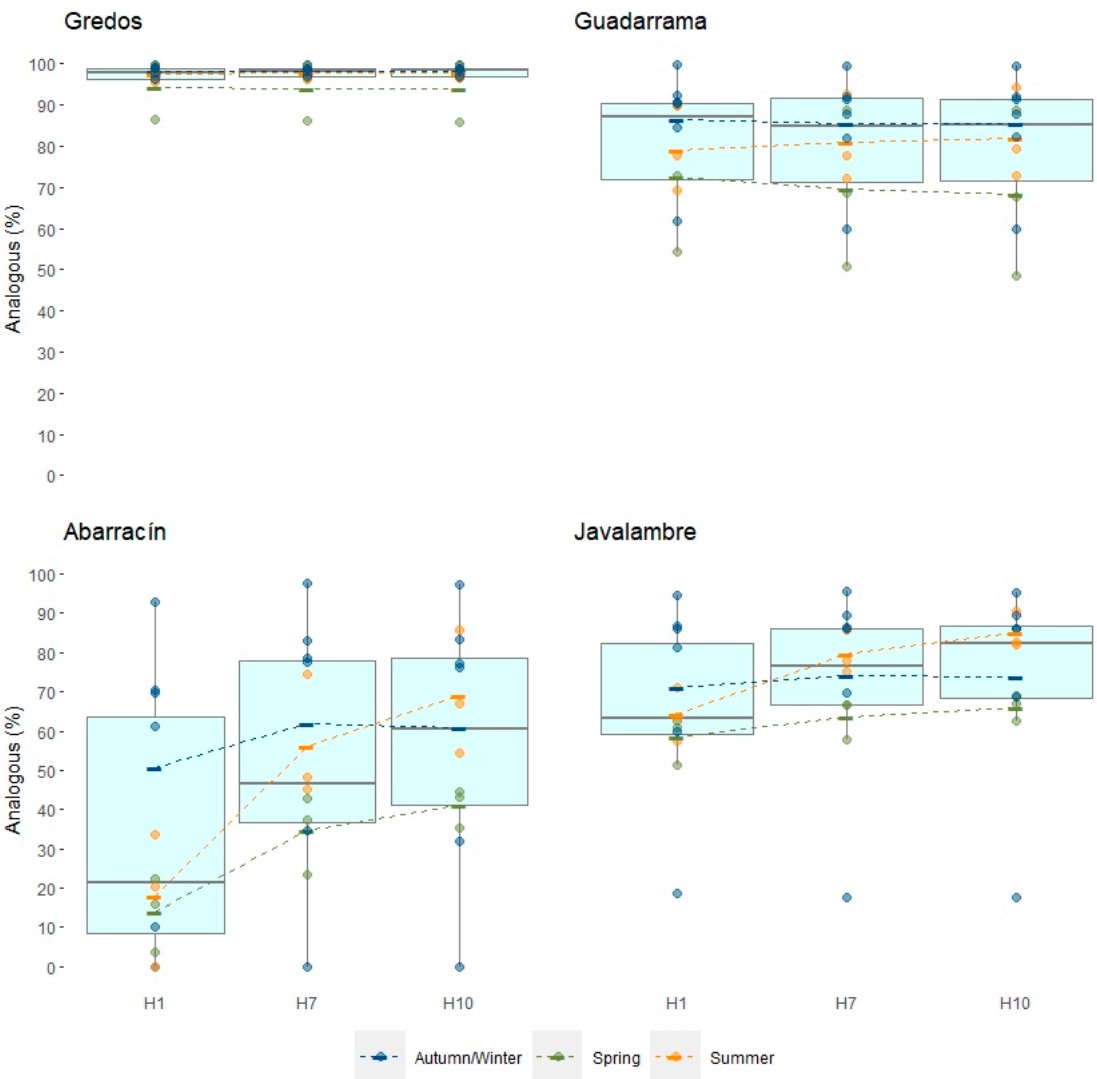

**Figure 6.** The percentage of 30 m pixels showing analogous modelled microclimate temperatures from 1980-89 to 2010-19 for three reference habitat types in the four focal mountain 6 × 6 km cells. Each panel shows one region, with data grouped by habitat type: pine forests "H1", scrublands "H7" and grasslands "H10". Coloured bars show mean for each season and habitat; habitats joined in each region by dotted lines. Box plots show median and interquartile range across all months in each habitat type.

## 4. Discussion

The ecological importance of mountain biodiversity and its potential vulnerability to climate change mean that reliable methods are needed to estimate climatic conditions in mountain regions. Our results indicate that mesoclimate data and a microclimate model can provide relatively accurate measures of spatial and temporal variation in temperatures experienced in a Mediterranean mountain range, but care is required in their interpretation and application to conservation. For example, absolute temperatures experienced near to the ground show considerable fine-resolution variation related to topography and vegetation, as demonstrated by our field-recorded temperatures [31]. Furthermore, we show that the spatial resolution and extent of climatic data, as well as the time of year and habitat of interest, have important implications for modelled exposure to changing climatic conditions. Scientists and environmental managers seeking to assess the vulnerability to climate change of mountain taxa or ecosystems, and to prioritize climate change refugia for conservation, therefore need to interpret climatic data at spatial scales that are appro-

priate for the valued conservation resource or associated management decision that are being assessed.

### 4.1. Field Validation of Microclimate Model

Mechanistic models have recently been developed that permit fine-resolution estimates of microclimate for regions without detailed coverage of meteorological stations [28,29], and their use has been validated for remote and topographically variable environments [30]. We show for a mountain ridge in the central Iberian Peninsula that temperatures estimated at 30 m resolution using the model Microclima led to a wider range of monthly maxima than 1 km resolution mesoclimate data, and the microclimate model also showed lower absolute differences than mesoclimate data from field measurements. The modelled temperature range was not as wide as that recorded by dataloggers at ground level, with measured maxima at ground level exceeding modelled temperatures, especially during summer (Figure 4). There can be large errors in the empirical measurement of temperatures in sunny microenvironments (such as at ground level in summer), which make it difficult to draw firm conclusions from direct comparisons between observed and modelled temperatures [44]. In this respect, results from mechanistic models are less affected than field data by fine-scale variation in factors such as shading and vegetation height, which are challenging to control for in-field assessments of microclimate [44]. Thus, the microclimate model could be a useful tool to estimate the relative effects of temporal and fine-resolution spatial variation in topography on the temperatures experienced by organisms near the ground, with the proviso that temperatures experienced are highly sensitive to very fine-scale variation in microtopography and vegetation, and that the models could underestimate absolute exposure to extreme maximum temperatures.

The application of the microclimate model for estimating minima again led to close correlations with observed variation in temperature over space and time, but led to substantial underestimates of the absolute values (Figure 4). Our measurements of field minimum temperature seldom went below zero, likely because of insulation by snow cover in winter, and the buffering effects of shrub cover and leaf litter on nighttime temperatures [39]. In contrast, the range and absolute values of monthly minima from the mesoclimate data were very close to those recorded by the field measurements, although the coarse resolution (1 km) of these data would preclude their application for estimating fine-resolution variation in minima. For study organisms in which fine-spatial-resolution variation in minima are important, in situ reference data can be used to help parameterize the microclimate model. Recent developments to the Microclima model have incorporated the effects of soil moisture, and ongoing refinements are being implemented to account for the effects of snow cover on temperature [30] (I.M.D. Maclean, *pers. comm.*).

### 4.2. Exposure to Non-Analogous Temperatures over Time

One method to estimate the exposure of organisms or ecosystems to climate change is to consider the proportion of the landscape in which conditions present in one period are maintained in another (analogous), versus locations where conditions are lost from the landscape or novel conditions appear (non-analogous). Based on monthly temperature maxima, we found differences in exposure to non-analogous conditions depending on the mountain region, the scale of analysis, the season of the year, and habitat type. Understanding how these effects operate could be important to inform assessments of climate change vulnerability for different species or locations.

Climatic conditions and rates of climate change are heterogeneous over space, and are influenced at meso- and microscales by factors such as regional weather patterns, elevation, distance to the coast, geology, and topographic factors such as shading, aspect and slope [12,19]. In our results, the study region with the narrowest elevation range (Albarracín, Figure 1) showed low proportions of analogous temperatures between 1980-89 and 2010-19 at both meso- (1 km resolution in 50 km squares) and microscales (30 m resolution in 6 km squares), despite having considerable internal topographic variation.

We interpret that the changes to broader regional conditions in some months were so marked that these overwhelmed the capacity of local microclimatic variation as a buffer. It is interesting to note that in some months for locations in this region, temperature decreases between the two decades led to monthly maxima that were below the lowest maxima recorded in 1980-89.

The scale of analysis had a clear influence on detection of non-analogous conditions: in the central 6 × 6 km cells used for microclimate modelling, both lost and gained non-analogous temperatures were detected at 30 m resolution, but were missed using data at 1 km resolution (see Figure 4 for Sierra de Albarracín). Greater climatic variability is detected at the fine scale due to the effects of topographic variation on physical processes of air movement and solar radiation [14,24,25]. The proportion of non-analogous conditions increased at the microscale compared with the mesoscale in three of the regions (Guadarrama, Albarracín and Javalambre), but not in Gredos. In contrast, in Gredos, expanding the assessment to a 50 × 50 km cell at 1 km resolution (mesoscale) led to increased estimates of exposure to non-analogous temperatures. Gredos was the region with the broadest elevation range across the mesoscale, with a ridge and plateau in the northern half and lower elevation plains in the south (Figure 1), and at the microscale it also showed a steep elevation gradient with relatively little internal topographic variability (largely a system of steep south-facing slopes rising to high elevations). Warming therefore led to a relatively uniform mesoscale shift, with novel hot conditions gained in the low-elevation southern half of the square, but cool conditions mainly maintained across high elevations (Figure 4). The regional topographic context, alongside broader prevailing changes to the climate, therefore influences the scale dependence of exposure to non-analogous conditions.

Rates of climate change have varied throughout the year in the Iberian Peninsula [11], and we recorded greater exposure to non-analogous temperatures in spring and summer than in autumn and winter (Figure 5). Understanding seasonal variation in climate change exposure is important because different behaviours and stages of growth or life cycles are sensitive to climatic conditions at different times of year. For example, the phenology of many butterfly species in central Spain is most sensitive to temperature variation in spring, immediately before annual adult emergence [45]. Shifts in phenology, alongside local shifts in distribution, could compensate for the monthly exposure to non-analogous climates that we have identified. However, most butterfly species analysed in north-east Spain have not shifted their flight periods since the 1990s because of cooling temperatures during the months that are critical for phenological sensitivity [41].

*4.3. Implications for Adapting Conservation to Climate Change*

Observed or expected exposure to climate change represents an important component of climate change vulnerability assessments for prioritizing conservation among biota and geographic locations [6]. The highly idiosyncratic responses to climate change that have been shown by co-occurring species [46] could result partly from the fact that, as we demonstrate, exposure to non-analogous conditions varies markedly among regions, seasons of the year, habitat types and spatial scales. Climate change impact assessments at coarse scales (>10 km resolution), such as in most bioclimate models to date [7], might give a broad idea of changing climatic favourability (e.g., [47]) but will miss important differences between taxa and geographic locations at the landscape scales that are important in planning and managing in situ conservation [48].

We show that there has been marked local variation in exposure to changing temperatures in the past three decades within and among mountain landscapes at both 1 km and 30 m resolution. Evidence of how exposure varies at these scales can be employed to help understand ecological responses and adapt management decisions to climate change. Across a mountain landscape, mesoclimatic (100 m–1 km) information allows for the identification of sites (e.g., systems of ridges or valleys) where topography is most likely to buffer the effects of climate change and therefore enable the persistence of metapopulations of threatened insects, or home ranges of more wide-ranging organisms such as mammals or

birds. Within these landscapes, microclimatic models (<100 m) permit the identification of specific topographic features (e.g., hollows, talus slopes) that provide isolated or unique conditions. Such locations may be targeted for ecological surveying for threatened taxa, or habitat management to maintain suitable biotic conditions, and following such approaches for monitoring to test whether in situ management has allowed population persistence or recovery. Plants and sedentary invertebrates may be able to adjust their distributions or behavioural thermoregulation at the scales where microclimate models can now assess exposure, and monitoring such responses for highly threatened species can help conservation managers to assess the efficacy of ongoing conservation adaptation to climate change.

Our results indicate that estimates of climate change exposure are scale-dependent. Therefore, the scale of analysis to inform vulnerability assessments depends in part on the focal "valued resource" [6]. However, complementary climate change exposure assessments at more than one spatial scale could be especially valuable when planning management for protected areas or regions in which multiple threatened species or communities—or indeed multiple metapopulations of focal species—occur [32], combined with validating refugial predictions using empirical ecological, genetic or physiological data [49]. In regions such as mountains with high levels of mesoclimatic and microclimatic variation, these assessments could allow important zones to be highlighted, and within these, microrefugial locations to be pinpointed for the protection and management of localized populations threatened by climate change.

## 5. Conclusions

Relatively accurate sources of mesoclimatic and microclimatic data are now available to fulfil the urgent need to assess climate change exposure in mountain regions. These models can reduce the investment of time and resources in temperature-recording devices in regions lacking in meteorological stations, and this work shows how the models can be validated using dataloggers in focal sites to support their application [28,30,44]. Our results for Mediterranean mountains in the Iberian peninsula show that measuring and modelling exposure at these complementary spatial scales can provide important information to understand ecological responses to climate change. Above all, we recommend that planners and managers conduct exposure assessments at scales that are relevant to organisms, ecosystems and their conservation management in a changing climate.

**Supplementary Materials:** The following supporting information can be downloaded at: https://www.mdpi.com/article/10.3390/land11112052/s1, The supplementary material files include data presented in the work, with Supplementary Metadata file to explain data in the remaining datafiles. Supplementary File 1 contains the temperature maxima and minima from dataloggers used to validate the associated mesoclimatic and microclimatic data. Supplementary File 2 contains the information on the number of gained, lost and analogue conditions corresponding to the mesoclimatic and microclimatic data for the four regions. Supplementary File 3 contains the information utilized to model the differences in analogue conditions between scales. Supplementary File 4 contains the data to model the differences in analogue conditions between habitats. This supplementary material is intended to permit reproducibility of the results presented.

**Author Contributions:** Conceptualization, R.J.W., M.M., G.U. and M.G.-V.; methodology, R.J.W., M.M., G.U. and M.G.-V.; formal analysis, M.M., G.U. and M.G.-V.; investigation, R.J.W., M.M., G.U. and M.G.-V.; resources, R.J.W.; data curation, M.M., G.U. and M.G.-V.; writing—original draft preparation, M.G.-V., M.M. and G.U.; writing—review and editing, R.J.W.; visualization, M.M., G.U. and M.G.-V.; supervision, R.J.W.; project administration, R.J.W.; funding acquisition, R.J.W. All authors have read and agreed to the published version of the manuscript.

**Funding:** This research was funded by the Spanish Ministerio de Ciencia, Innovación y Universidades (MCIU), the Agencia Estatal de Investigación (AEI) and the EU Regional Development Fund (FEDER, UE), grant number RTI2018-096739-B-C21.

**Data Availability Statement:** Data supporting reported results are presented in the Supplementary Material.

**Acknowledgments:** Access and research permits for model validation in the Sierra de Guadarrama were provided by the Comunidad de Madrid, and David Gutiérrez assisted with fieldwork. Thank you to Ilya Maclean for advice about development and implementation of the Microclima model.

**Conflicts of Interest:** The authors declare no conflict of interest.

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
