# Peer review of "Assessing Climate Change Exposure for the Adaptation of Conservation Management: The Importance of Scale in Mountain Landscapes"

_land, doi:10.3390/land11112052_

Round 1

Reviewer 1 Report

Great paper.  Important contribution to this literature. I only have minor comments, save 1: Your reasoning (lines 174-176) makes it seem like it should have been 1970-1979, if you were choosing this period as a kind of a baseline. Also not clear why you used CHELSA starting at 1979, not 1980

-46: delete comma after "km)"

-64: citation 6 can also be cited here

-87: would be good to say a few more words about what analogous vs non-analogous climatic conditions means. This isn't explained until 193-201 (though there it is explained well!)

-137: add commas around 'with the resulting data'

Hope the next paper looks at validating the climate change refugia with species data, a la Barrows et al 2020! https://esajournals.onlinelibrary.wiley.com/doi/10.1002/fee.2205

Author Response

Reviewer 1

Great paper.  Important contribution to this literature.

Authors>>We sincerely acknowledge all these positive comments.

I only have minor comments, save 1: Your reasoning (lines 174-176) makes it seem like it should have been 1970-1979, if you were choosing this period as a kind of a baseline. Also not clear why you used CHELSA starting at 1979, not 1980

Authors>> Good point – we have clarified our reasoning on lines 183-188, and explained that the CHELSA data were only available from 1979 onwards. On line 196 we clarify that the CHELSA data we used were from 1980-89 and 2010-19.

Line 46: delete comma after "km)"

Authors>> Done (line 48)

Line 64: citation 6 can also be cited here

Authors>> Added (line 66), thanks for the suggestion.

Line 87: would be good to say a few more words about what analogous vs non-analogous climatic conditions means. This isn't explained until 193-201 (though there it is explained well!)

Authors>> Thanks for the suggestion. In this new version we added a description about analogous vs non-analogous climatic conditions in the Introduction on lines 89-95.

Line 137: add commas around 'with the resulting data'

Authors>> Added (line 146)

Hope the next paper looks at validating the climate change refugia with species data, a la Barrows et al 2020! https://esajournals.onlinelibrary.wiley.com/doi/10.1002/fee.2205

Authors>> Thank you very much for this suggestion for future work. We have added this next step and useful reference to line 496 of the Discussion.

Reviewer 2 Report

Comments on Gómez Vadillo et al., Assessing climate change exposure

Coarse-grained macroclimatic models are limited when it comes to predicting the fate of endemic species living in mountains in a restricted space and elevation zone.

The paper tests two mechanistic climate models at two spatial scales, one at a 1 km resolution, the other at a 30 m resolution by comparing model predictions to ground-level temperature measurements. 

The paper appears well-designed, well-written and important for the assessment of mountain species to climate change. However, both climate models are not very accurate in capturing temperature extremes, which are much more pronounced than in the model predictions. That seems to be a serious limitation, since local population dynamics may be determined by temperature extremes. A poor prediction of extremes may lead to an underestimation of environmental stochasticity, and consequently, extinction probability of local populations, when applied to climate change-related risk assessment of species.

Consequently, the conclusions in the abstract and in the conclusions should probably be toned down and put into perspective: The statement in line 269 (“For study organisms in which fine spatial resolution variation in minima are important, in situ reference data can be used to help parameterize the microclimate model, although at present Microclima is unable to account for the effects of snow cover on temperature [41]is an important and necessary suggestion how to improve the model predictions. The general importance of the mechanistic models has been demonstrated, and the general usefulness of the approach is justified, but the numerical disagreement of the model predictions and the recorded temperature measurements appear large and consequential. So maybe the authors could address these discrepancies in more details and suggest how to cope with them. Is there way to refine the models? 

Overall, the paper is an important step forward towards a better understanding of mountain microclimate and its implications on the distribution of species during climate change. It definitely should be published in the journal Land. However, the paper also illustrates the limitations of the models. As it seems from the results, some more model sophistication is still needed before reliable predictions on future climate change-triggered distributions of threatened mountain can be made.

Some remarks:

p. 7 Fig. 3: The colours are not explained.

p. 9 Fig. 4: Correct to “monthly maximum (above)” and “minimum (below)”

line 237: Maxima are not well covered in the models.

line 337: If the wrong model is selected, the differences to the recorded temperatures are substantial.

line 353: Not capturing the extremes seems to be problematic limitation to the microclimatic model

Author Response

Reviewer 2

Coarse-grained macroclimatic models are limited when it comes to predicting the fate of endemic species living in mountains in a restricted space and elevation zone.

The paper tests two mechanistic climate models at two spatial scales, one at a 1 km resolution, the other at a 30 m resolution by comparing model predictions to ground-level temperature measurements. 

The paper appears well-designed, well-written and important for the assessment of mountain species to climate change. However, both climate models are not very accurate in capturing temperature extremes, which are much more pronounced than in the model predictions. That seems to be a serious limitation, since local population dynamics may be determined by temperature extremes. A poor prediction of extremes may lead to an underestimation of environmental stochasticity, and consequently, extinction probability of local populations, when applied to climate change-related risk assessment of species.

Consequently, the conclusions in the abstract and in the conclusions should probably be toned down and put into perspective: The statement in line 371 (“For study organisms in which fine spatial resolution variation in minima are important, in situ reference data can be used to help parameterize the microclimate model, although at present Microclima is unable to account for the effects of snow cover on temperature [41]) is an important and necessary suggestion how to improve the model predictions. The general importance of the mechanistic models has been demonstrated, and the general usefulness of the approach is justified, but the numerical disagreement of the model predictions and the recorded temperature measurements appear large and consequential. So maybe the authors could address these discrepancies in more details and suggest how to cope with them. Is there way to refine the models? 

Overall, the paper is an important step forward towards a better understanding of mountain microclimate and its implications on the distribution of species during climate change. It definitely should be published in the journal Land. However, the paper also illustrates the limitations of the models. As it seems from the results, some more model sophistication is still needed before reliable predictions on future climate change-triggered distributions of threatened mountain can be made.

Authors>> Thank you very much for the positive assessment of our manuscript and for our demonstration of the general importance of mechanistic microclimate models in understanding ecological responses. We agree that a good understanding of the limitations and sophistication of such models is needed, and have therefore added provisos and toned down our language in several parts of the manuscript. In the Abstract (lines 16-17) we have noted the tendency for the models to somewhat underestimate observed ground-level maxima. We have also modified the final sentence of the Abstract to note that the models are “potentially” useful tools (line 25). We have similarly qualified the first paragraph of the Discussion (lines 358-363), added more nuanced discussion on lines 376-392, and the key point identified by the reviewer about refinements to the model on lines 406-409 (for which we contacted the author of the model, Dr Ilya Maclean – also now included in the Acknowledgments).

Below we detail our responses to the additional specific comments of reviewer 2.

Some remarks:

  1. 7 Fig. 3: The colours are not explained.

Authors>> In this new version, figure 3 colours have been explained in the figure legend.

  1. 9 Fig. 4: Correct to “monthly maximum (above)” and “minimum (below)”

Authors>> Modified.

Line 237: Maxima are not well covered in the models.

Authors>> We have addressed this point (and the wider comments of Reviewer 2) in several places. Please see our comments above about the additions we have made to the Abstract to tone down our conclusions (lines 16-17, 25). Specifically in the results, we have drawn explicit attention to the under-estimation of maxima (especially in summer) by extending the description of the results on lines 252-257: “However, the temperature range of monthly maxima was much greater for the field measurements (minimum -1.4, maximum 63.4 ºC), particularly during summer months (June-September) when there was little overlap of field-recorded monthly maxima with the microclimate model or, especially, the mesoclimate data (Figure 4).”

Line 337: If the wrong model is selected, the differences to the recorded temperatures are substantial.

Authors>> We have expanded the opening paragraph of the Discussion to improve our interpretation of the main results (lines 357-365): “Our results indicate that mesoclimate data and a microclimate model can provide relatively accurate measures of spatial and temporal variation in temperatures experienced in a Mediterranean mountain range, but that care is required in their interpretation and application to conservation. For example, absolute temperatures experienced near to the ground show considerable fine resolution variation related to topography and vegetation, as demonstrated by our field recorded temperatures [31]. Furthermore, we show that the spatial resolution and extent of climatic data, as well as the time of year and habitat of interest, have important implications for modelled exposure to changing climatic conditions.

Line 353: Not capturing the extremes seems to be problematic limitation to the microclimatic model

Authors>> We have extended the paragraph on lines 371-395 to properly nuance the applications and limitations of the models in the light of the results. We include reference to the limitations to the accuracy of field-measured extreme temperatures themselves, (reference 46, Maclean et al., 2021. On the measurement of microclimate. Methods in Ecology and Evolution, 12: 1397-1410. https://doi.org/10.1111/2041-210X.13627).

Reviewer 3 Report

See files

Author Response

Reviewer 3

Authors>> We are grateful for the overall assessment and review comments submitted by reviewer 3. We are glad again that the review felt that the scope, content and approach were appropriate, and we outline below how we have addressed the three specific recommendations for improvement given by the referee.

Referee 3 had two comments on the Abstract:

Line 13-15: by using what method or generated from exsiting models?

Line 22-25: The significance should be more specific

Authors>>In our revision we have taken on board these comments and those of Reviewer 2, making textual changes to the Abstract to address the recommendations whilst not exceeding the maximum word count. To address referee 3’s first comment, we have named the microclimate model and source of the mesoclimate data, and provided the references for these (line 14-15). To address the second comment and the general comments of Reviewer 2, we have specified more clearly throughout the Abstract when we are referring to temperature maxima and the possibility that the models may somewhat underestimate maximum ground level temperatures. We have also rewritten the final sentence of the Abstract to make a more specific (yet nuanced, considering Reviewer 2’s comments) final statement (line 25).

The other comment referred more generally to the Materials and Methods:

Line 102: Please specify the method you used and generate an analytical framework for audience to understand more clearly.

Authors>>In response to one of Reviewer 1’s comments (above), we added several lines explaining the rationale behind our assessment of analogous and non-analogous temperatures, in the last paragraph of the Introduction (lines 89-95). In this “objectives” paragraph we explain clearly our approach: first ground-truthing the microclimate and mesoclimate models (lines 84-86), then applying them across replicate mountain landscapes to assess recent exposure to climate change (lines 87-99), and then contextualising them in terms of the types of organism or management decision that are likely to be affected depending on the scale of analysis (lines 99-100). We hope that this revised section provides the necessary preface to the Materials and Methods to help readers understand and evaluate our approach.